# ReTool: Reinforcement Learning for Strategic Tool Use in LLMs

**Jiazhan Feng**[1][*], **Shijue Huang**[1,2][*][†], **Xingwei Qu**[1], **Ge Zhang**[1], **Yujia Qin**[1],
**Baoquan Zhong**[1], **Chengquan Jiang**[1], **Jinxin Chi**[1], **Wanjun Zhong**[1][‡]

[1]ByteDance Seed  [2]The Hong Kong University of Science and Technology
{fengjiazhan,wanjun}@bytedance.com joehsj310@gmail.com

## ABSTRACT

While reasoning models trained with reinforcement learning (RL) excel in reasoning, they struggle in scenarios requiring structured problem-solving, such as geometric reasoning, concise computation, or complex equation solving—areas where computational tools like code interpreters (CI) demonstrate distinct advantages. To bridge this gap, we propose **ReTool**, which enhances long-form reasoning with tool-integrated learning, including two key features: (1) dynamic interleaving of real-time code execution within natural language reasoning processes, and (2) an automated RL paradigm that allows policy rollouts with multi-turn real-time code execution and teaches the model in learning when and how to invoke tools based on outcome feedback. ReTool employs a systematic training framework, beginning with synthetic code-augmented long-form reasoning data for cold-start training. Subsequent RL training leverages task outcomes as rewards to iteratively refine the model's tool use strategy, enabling autonomous discovery of optimal tool invocation patterns without human priors. Experiments on challenging MATH Olympiad benchmark AIME demonstrate ReTool's superiority: Our 32B model achieves 67% accuracy with 400 training steps, outperforming text-based RL baseline (40% accuracy, 1080 steps) in performance and efficiency. Remarkably, ReTool-32B attains 72.5% accuracy in extended settings, surpassing OpenAI's o1-preview by 27.9%. Further analysis reveals generalization to broader tool-use scenarios and emergent behaviors such as code self-correction, signaling an "aha moment" in which the model autonomously masters adaptive tool use. These findings highlight the promise of outcome-driven tool integration for advancing complex mathematical reasoning and offer new insights into hybrid neuro-symbolic systems.

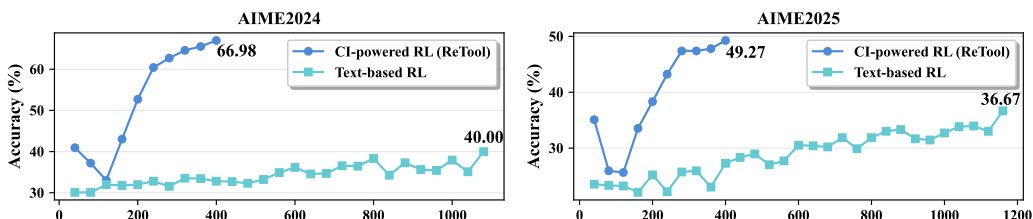

Figure 1: AIME scores of ReTool and text-based RL baseline based on the Qwen2.5-32B-Instruct model.

---

[*]Equal contributions.

[†]This work was done during an internship at Bytedance Seed.

[‡]Corresponding author.

# 1 INTRODUCTION

Reinforcement learning (RL) has recently become a popular paradigm for enhancing the reasoning capabilities of large language models (LLMs), enabling them to explore and refine long chains of thought (CoT) (Wei et al., 2022; Yao et al., 2023; Luong et al., 2024; Zhang et al., 2024). Reasoning models such as OpenAI o1 (OpenAI et al., 2024) and DeepSeek R1 (DeepSeek-AI et al., 2025) demonstrate strong performance in pure text-based reasoning tasks by learning to self-correct and engage in more deliberate, analytical thinking (Claude, 2025; Team, 2025d;a). These advances suggest early signs of metacognitive control, where models not only reason, but also monitor and revise their reasoning process.

Despite these advances, reasoning LLMs equipped with long chains of textual reasoning processes (Ouyang et al., 2022) still show notable limitations in tasks that require precise numerical calculation or symbolic manipulation, such as geometric reasoning, precise computation, or complex equation solving. In contrast, computational tools, such as code interpreters (CI), can empower models with symbolic computation capabilities that go far beyond pure text-based reasoning. Unlike textual CoT (Wei et al., 2023) methods that rely solely on internal language patterns, code interpreters provide a formal and executable interface for enumeration, verification, and precise computation. This not only enables exact numeric validation of intermediate steps—dramatically reducing the ambiguity and compounding error often seen in textual reasoning (Chen et al., 2023; Wang et al., 2023b), but also allows models to expand their solution search space via programmable exploration.

Recent works have explored prompting and supervised fine-tuning methods (Chen et al., 2025b; Pan et al., 2023) to equip LLMs with tool-use capabilities. However, these approaches are limited to imitating the specifically-curated data distribution, often failing to generalize beyond seen patterns or adaptively decide when and how to invoke external tools. As a result, models may misuse tools or fall back on brittle heuristics that are not robust across diverse problem settings. To overcome these limitations, RL offers a principled solution: it enables models to explore flexible reasoning trajectories and learn tool-use strategies guided by outcome-based feedback. This paradigm not only incentivizes correct solutions, but also allows the model to discover nuanced behavioral patterns—such as how to recover from tool execution mistakes via self-correction, decide when to effectively invoke tool execution during the long-chain reasoning process.

In this work, we embrace the RL paradigm and introduce **ReTool**, a **Tool**-augmented **Re**inforcement learning framework explicitly designed to guide LLMs towards optimal strategies for leveraging external computational tools during reasoning. ReTool consists of two key components: First, we develop a data construction pipeline to curate a high-quality cold-start dataset that explicitly demonstrates when and how to invoke the code interpreter. This teaches the model an initial competency in tool usage and execution result analysis. Then, we apply tool-enhanced reinforcement learning to train the model in discovering optimal tool manipulation reasoning strategy and adjusting its behavior through outcome-based rewards, going beyond what can be captured by supervised learning alone. During long-chain reasoning, the policy model rolls out by flexibly writing code blocks and achieving real-time execution results from a sandbox-style code interpreter to assist subsequent thinking.

We evaluate ReTool on the challenging MATH Olympiad benchmarks AIME2024 and AIME2025. Building on Qwen2.5-32B-Instruct (Yang et al., 2024a), our model achieves 67.0% accuracy on AIME2024 with only 400 training steps, significantly outperforming the text-based RL baseline, which achieves 40.0% accuracy with 1080 training steps. These substantial gains highlight that explicitly modeling tool-use as part of the decision process not only pushes limits of model reasoning but also enhances training efficiency. Furthermore, when trained on DeepSeek-R1-Distill-Qwen-32B (DeepSeek-AI et al., 2025), our model demonstrates further improvements, surpassing competitive baselines such as QwQ-32B-Preview (Team, 2025d), s1-32B (Muennighoff et al., 2025), and OpenAI o1-preview (OpenAI, 2024). This suggests that the RL training process inspires more efficient problem-solving strategies. We further conduct a comprehensive analysis, including ablation study, extension to the web search domain, CI cognitive behavior through RL training. This analysis highlights several key findings: our model demonstrates enhanced code utilization capabilities, enabling it to employ more accurate and complex code snippets; It also learns to invoke tools appropriately, select tool adaptively, structure tool calls effectively, and iteratively refine reasoning

through emergent code self-correction capabilities. Overall, our main contributions are summarized as follows:

- We propose **ReTool**, a novel reinforcement learning framework that integrates code interpreter execution into the reasoning loop of LLMs. To equip the model with foundational capabilities for invoking the code interpreter, we curate a high-quality cold-start dataset through our developed pipeline. Furthermore, we design a reinforcement learning framework that supports interleaved code execution during rollout, enabling the model to iteratively explore, refine, and optimize its reasoning strategies through tool-augmented interactions guided by feedback from a sandboxed code interpreter.

- As shown in section 3.6, we conduct comprehensive empirical and behavioral analyses, and observe several key findings: (1) After RL training, the response length is reduced by approximately 40% compared to that prior to training, showcasing the potential reasoning token efficiency of tool-integrated reasoning; (2) Our approach can generalize to broader tool-use scenarios like web search; (3) During RL training, the code ratio, code lines and correct code counts show increase trends, and the code invocation timing becoming shifts earlier, indicating the improved code use capabilities and strategic tool usage development; (4) Emergent behaviors like code self-correction and adaptive tool selection can be observed during RL phase, bringing more advanced tool-augmented reasoning patterns.

## 2 METHODOLOGY

In this section, we introduce ReTool. We begin with an overview of the framework, followed by a description of the cold-start training process, including the data construction pipeline and supervised fine-tuning (section 2.2). We then outline our reinforcement learning pipeline, enhanced by a code interpreter sandbox, to further enhance strategic tool usage development (section 2.3).

### 2.1 OVERVIEW

Our approach consists of two stages: cold-start supervised fine-tuning followed by reinforcement learning with interleaved code execution rollout. Firstly, we collect data through our designed pipeline for cold-start supervised fine-tuning (SFT), which provides a robust initialization for the reinforcement learning (RL) phase. To enhance our model's tool utilization capabilities, we introduce a specialized tool-using RL pipeline that enhances the model's ability to appropriately select and apply tools during the reasoning process.

### 2.2 COLD-START FOR TOOL-INTEGRATED REASONING FOUNDATION

We designed a pipeline to collect and curate high-quality data. Specifically, we begin by gathering existing mathematical reasoning data from diverse sources, including open-source datasets such as Open-Thoughts (Team, 2025c). Subsequently, we implement a dual-verification approach combining human expert curation and Deepseek-R1 (DeepSeek-AI et al., 2025) evaluation to filter invalid data. Through these steps, we collect a high-quality text-based reasoning dataset, denoted as $\mathcal{D}_{\text{init}}$.

Based on $\mathcal{D}_{\text{init}}$, we further construct code-integrated reasoning data automatically. We first utilize a structured prompt template (detailed in Appendix Figure 9) for transformation, which modifies the original thinking process by replacing manual calculation steps that can benefit from code execution with the corresponding code snippets and their interpreter's execution results. Following this initial transformation, we apply a two-stage verification protocol. The first stage focuses on format verification, which improves readability and ensures consistent syntax that that enables the efficient detection of computational tool invocation triggers during subsequent reinforcement learning phases. The second stage entails answer verification, where we eliminate data samples whose final outputs do not align with the correct solutions to the mathematical problems. Finally, we collect a dataset $\mathcal{D}_{\text{CI}}$ that consist of code-augmented long-form reasoning traces.

ReTool employs supervised fine-tuning to learn when and how to invoke the code interpreter from the aforementioned dataset $\mathcal{D}_{\text{CI}}$, thereby enhancing model's capability to appropriately utilize computational tools.

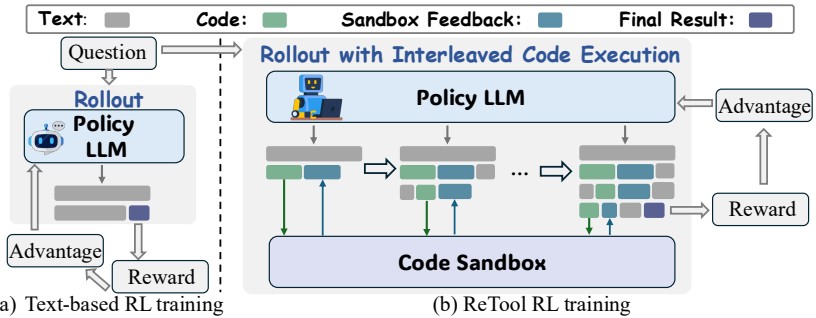

Figure 2: Demonstration of text-based RL training process and ReTool's RL training process.

## 2.3 ReTool: Reinforcement Learning for Strategic Tool Use

### 2.3.1 Training Algorithm

We train ReTool based on PPO (Schulman et al., 2017), it updates policy with the following objective:

$$\mathcal{J}_{\text{PPO}}(\theta) = \mathbb{E}_{(q,a)\sim\mathcal{D},o_{\leq t}\sim\pi_{\theta_{\text{old}}}(\cdot|q)} \left[ \min\left( \frac{\pi_\theta(o_t \mid q, o_{<t}; \mathcal{CI})}{\pi_{\theta_{\text{old}}}(o_t \mid q, o_{<t}; \mathcal{CI})} \hat{A}_t, \right.\right.$$
$$\left.\left. \text{clip}\left( \frac{\pi_\theta(o_t \mid q, o_{<t}; \mathcal{CI})}{\pi_{\theta_{\text{old}}}(o_t \mid q, o_{<t}; \mathcal{CI})}, 1 - \varepsilon, 1 + \varepsilon \right) \hat{A}_t \right) \right], \quad (1)$$

where $\pi_\theta$ is policy model, $\pi_{\theta_{\text{old}}}$ is reference model, $\pi_\theta(o_t \mid q, o_{<t}; \mathcal{CI})$ represents the rollouts with interleaved code execution and feedback from code interpreter. We modify PPO to better adopt tool integrated reasoning. During training, the policy LLM will collaborate with a code sandbox to generate rollouts with multi-turn real-time code execution for solving given problems. We implement a rule-based outcome reward to enable the model with the flexibility to autonomously explore and develop strategies for code usage awareness, code selection, timing of code invocation, and further diverse behaviors.

**Reward Design** To teach the model in learning when and how to invoke tools, we implement a rule-based accuracy reward to optimize the model. The accuracy reward evaluates response correctness. We require the model to present final answers in a specified format (e.g., within \boxed{}), enabling reliable rule-based verification. The reward is formulated as:

$$R(a, \hat{a}) = \begin{cases} 1, & \texttt{is\_equivalent}(a, \hat{a}) \\ -1, & \text{otherwise} \end{cases} \quad (2)$$

where $a$ and $\hat{a}$ represent the ground-truth answer and the predicted answer, respectively. We simplify the reward design aim to alleviate reward hacking and promote more diverse problem-solving behaviors based on mere outcome feedback without considering code executability reward.

**Rollout with Interleaved Code Execution** To facilitate the integration of reasoning and executable code within the model, we propose a rollout approach that dynamically supports interleaved real-time code execution with natural language reasoning processes. As depicted in Figure 2 (b), our rollout process differs from the conventional approach, which typically generates only text-based reasoning (as shown in Figure 2 (a)). By contrast, our rollout approach integrates the collaboration of a policy LLM with an external code sandbox, enabling the production of hybrid content that combines text, code snippets, and real-time interpreter feedback. Concretely, we utilize a prompt template (Appendix Figure 8) to guide the model in interacting with the code sandbox by utilizing tags  to explicitly mark the boundaries of generated codes. During the rollout process, policy model generate text-based reasoning $t_1$ when a code termination trigger () is detected, the generation pause and the generated code $c_1$ is parsed and send to code sandbox environment for execution. Upon completion, the sandbox's output $f_1$ (successful results or error messages) is filled within <interpreter></interpreter> tags and fed back to the model, which continues generating the rollout until either providing a final answer $o$ or producing a new code snippet, ultimately producing a hybrid reasoning trajectory $[t_1 \oplus c_1 \oplus f_1 \oplus ... \oplus o]$.

Notably, our approach returns both successful code execution results and interpreter error messages to the model. This dynamic feedback mechanism enables the model to iteratively explore, refine, and optimize its reasoning and tool usage strategies.

### 2.3.2 TRAINING DETAILS

**Cold-start & RL** For training, we employ the VeRL framework . We adopt PPO as our RL method. We train our model on curated cold-start data for two epochs. Regarding hyperparameters, we utilize the AdamW optimizer with an initial learning rate of 1e-6. We define the expected maximum sequence length as 16384 tokens. For training, the mini-batch size is set to 512, and the KL coefficient is set to 0.0. We use Qwen2.5-32B-Instruct (Qwen et al., 2025) as the main backbone. All experiments are conducted on NVIDIA H20 GPUs.

**Interpreter Feedback Mask.** We mask out the `<interpreter></interpreter>` feedback output from the loss computation. This sandbox-based output masking approach blocks external tokens from interfering with loss calculations, ensuring training stability and preserving the model's inherently generated coherent reasoning sequences from disruption.

**KV-Cache Reuse.** In order to reduce the memory cost during rollout, when each time the code termination trigger (``) is detected, we will cache all the KV-cache before code execution and only calculate and append the KV-cache from the interpreter feedback (`<interpreter></interpreter>`). This will largely reduce the KV-cache for each rollout.

**Sandbox Construction.** To accelerate the RL training process, we design a asychornous code sandbox environment. The sandbox pods function as workers in a pool, independently pulling tasks based on their current capacity, creating an efficient load-balancing mechanism. This distributed asynchronous approach accelerates RL training by enabling parallel environment interactions across multiple threads, It prevents slower threads from creating bottlenecks and ensures optimal resource utilization, maintaining continuous throughput during the training process.

## 3 EXPERIMENT

### 3.1 EVALUATION SETUP

To ensure a stable evaluation, we repeat the evaluation set AIME2024 and AIME2025 32 times, GPQA (Diamond) (Rein et al., 2023) 8 times, MATH500 (Hendrycks et al., 2021) 4 times, GSM8K (Cobbe et al., 2021) 2 times, and report the overall average accuracy to estimate pass@1. The inference hyperparameters of evaluation are set to temperature 1.0 and top-p 0.7. We compare ReTool with competitive baselines, including Qwen2.5-Math-72B-Instruct (Yang et al., 2024b), Qwen2.5-Math-72B-Instruct-TIR (Yang et al., 2024b), Sky-T1 (Team, 2025b), DeepSeek-R1-Zero-Qwen-32B (DeepSeek-AI et al., 2025), QwQ-32B-Preview (Team, 2025d), s1-32B (Muennighoff et al., 2025), OpenAI o1-preview (OpenAI, 2024). To verify the effectiveness of our ReTool, we also compare the performance with RL without tool-using, i.e. Text-based RL (Qwen2.5-32B-Instruct). And for the results of baselines, we report the avg@k by coping from corresponding literature source as pass@1.

### 3.2 MAIN RESULTS

As shown in Table 1, ReTool enables the LLM to flexibly leverage the code interpreter during the RL stage, leading to substantial performance improvements. Specifically, ReTool (Qwen2.5-32B-Instruct) achieves accuracies of 67.0% on AIME2024, 49.3% on AIME2025, 95.9% on GSM8K, 93.1% on MATH, and 58.7% on GPQA with only 400 training steps. This markedly outperforms most strong baselines with much larger parameter sizes, such as Qwen2.5-Math-72B. These findings indicate that the tool-integrated learning paradigm employed by ReTool enhances the model's reasoning capabilities significantly. Furthermore, on AIME2024, ReTool (Qwen2.5-32B-Instruct) surpasses the competitive baseline s1-32B by 10.3%. Similarly, on AIME2025, it achieves an 11.4% gain over OpenAI's o1-preview. When combined with a more advanced backbone, ReTool (DeepSeek-R1-Distill-Qwen-32B) further improves performance, achieving scores of 72.5% on AIME2024 and 54.3% on AIME2025. These results suggest that more effective problem-solving strategies are discovered during the RL training process.

Table 1: Main results on benchmarks in mathematic and STEM domains.

| Model | AIME2024 | AIME2025 | GSM8K | MATH500 | GPQA |
|---|---|---|---|---|---|
| Qwen2.5-Math-72B-Instruct | 30.0 | - | 95.9 | 85.9 | - |
| Qwen2.5-Math-72B-Instruct-TIR | 40.0 | - | 95.8 | 88.1 | - |
| Sky-T1 | 43.3 | - | - | 82.4 | 56.8 |
| OpenAI o1-preview | 44.6 | 37.9 | - | 85.5 | **73.3** |
| DeepSeek-R1-Zero-Qwen-32B | 47.0 | - | - | 94.3 | 62.1 |
| QWQ-32B-Preview | 50.0 | 33.5 | - | 90.6 | 54.5 |
| s1-32B | 56.7 | - | - | 93.0 | 59.6 |
| ReTool (Qwen2.5-32B-Instruct) | 67.0 | 49.3 | 95.9 | 93.1 | 58.7 |
| ReTool (DeepSeek-R1-Distill-Qwen-32B) | **72.5** | **54.3** | **96.3** | **95.2** | 62.3 |

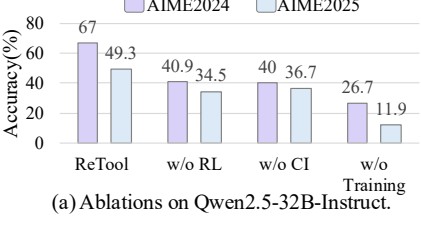

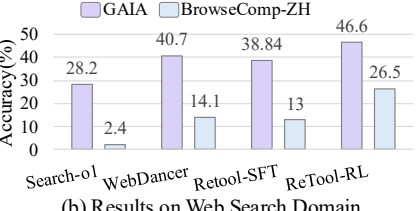

(a) Ablations on Qwen2.5-32B-Instruct.  (b) Results on Web Search Domain.

Figure 3: (a) Ablation study;(b) Results on the web search domain with same backbone on Qwen2.5-32B.

## 3.3 ABLATION STUDY

To further assess the effectiveness of ReTool, we conduct an ablation study and compare against several variants built on Qwen2.5-32B-Instruct: (1) w/o RL: the cold-start model that still incorporates the code interpreter; (2) w/o CI: a text-based RL method initialized with a text-only cold-start SFT to ensure fairness; (3) w/o Training: the original base model. As illustrated in Figure 3 (a), removing either the RL stage (w/o RL) or the CI integration (w/o CI) leads to a notable drop in performance on AIME2024&2025. Furthermore, our cold-start model achieves 40.9% accuracy on AIME2024, which is comparable to the text-based RL baseline (40.0%) and significantly higher than the base model (26.7%). These findings demonstrate that our curated dataset successfully captures tool-usage patterns within executable reasoning traces, and that CI-integrated training further enhances reasoning performance.

## 3.4 EXPANSION TO WEB SEARCH SCENARIO

To assess whether our strategy generalizes to broader tool-use scenarios, we further conducted experiments in the Web Search domain. Specifically, we applied ReTool with the Bing Search tool (following the MCP tool-use definition) on two widely used benchmarks: GAIA (Mialon et al., 2023) and BrowseComp-ZH (Zhou et al., 2025). As illustrated in Figure 3(b), when using the same backbone (Qwen2.5-32B), ReTool consistently outperforms competitive baselines built on the same backbone, including WebDancer (Wu et al., 2025) and Search-o1 (Li et al., 2025a). These findings highlight ReTool's generalizability and effectiveness beyond mathematical reasoning tasks, underscoring its adaptability across diverse tool-use environments.

## 3.5 "AHA MOMENT" OF CODE SELF-CORRECTION.

Interestingly, our model exhibits an emergent ability to self-correct non-executable code, despite the absence of explicit training data for code self-correction. As shown in Figure 6 in Appendix, the model initially produced code that failed to execute due to the undefined function "greedy()". Upon receiving feedback from the interpreter, the model recognized the error and responded with the reflection: "**Oops, the functions need to be defined in the same scope. Let's correct that**." It then proceeded to generate a revised, executable version of the code that included all necessary function definitions. This emergent behavior suggests that reinforcement learning can foster metacognitive

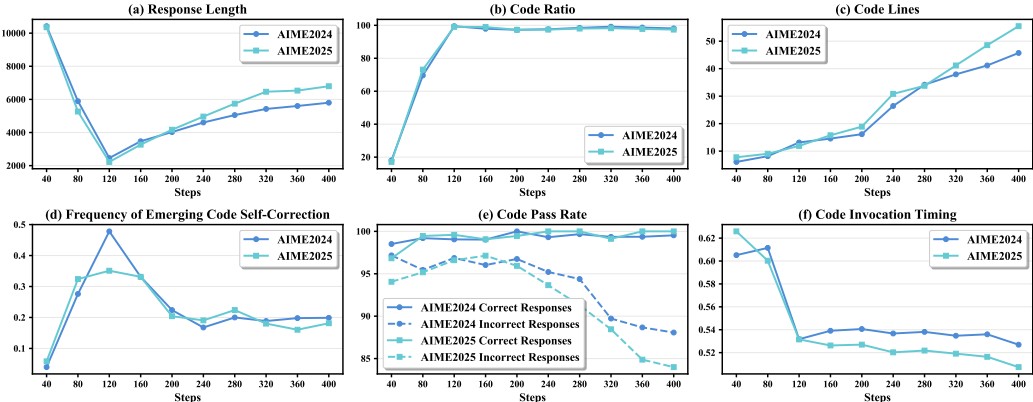

Figure 4: CI-related behavior evolution during RL training.

capabilities, enabling the model to iteratively refine its generated code to address more complex problems.

## 3.6 COGNITIVE ANALYSIS

We present a comprehensive analysis, including the dynamics of code interpreter (CI)-related behaviors throughout the RL and the differences in code purpose before and after RL.

**CI-related Behavior Evolution.** To gain deeper insights into the RL process of ReTool, we systematically evaluated CI-related metrics. Specifically, we computed various metrics by analyzing model-generated outputs on the AIME2024 and AIME2025 datasets based on each saved checkpoint during RL training:

- **Response Length** (Figure 4 (a)): We calculated the average response length and observed a distinct trend: the response length initially declines sharply, later followed by a relatively gentle increase. We attribute the initial decline to the replacement of complex computational processes with more concise code, while the subsequent rise is likely due to the emergence of more diverse and complex code behaviors during RL training. Notably, the final average response length remains 40% shorter than that before RL training (i.e., from 10k to 6k). This suggests that the CI-powered reasoning approach potentially enhances efficiency of reasoning token utilization ratio by replacing intricate computational processes with code.

- **Code Ratio** (Figure 4 (b)): The ratio of responses that contain code are also calculated. Analysis reveals that throughout the RL training process, the average code ratios exhibit a total upward trend and end with covering nearly 98% percent of all questions. This suggests that the model's proficiency in code utilization improved progressively during the RL process, facilitating strategic tool usage development.

- **Code Lines** (Figure 4 (c)): The lines of generated code reflects its complexity to some extent. Observations show that the average code lines in responses exhibits a consistent upward trend throughout training. By the end of RL training, the final average code lines is nearly fivefold higher than that before RL training. This trend suggests that the model has learned more complex code strategies during the RL phase.

- **Frequency of Emerging Code Self-Correction** (Figure 4 (d)): We approximate the frequency of code self-correction by detecting specific turning cues (e.g., oops, wait, or correcting) that appear between a failed code block and its immediate successor. As shown, the frequency of self-correction emerges early and peaks during the initial stages of training, then gradually decreases and stabilizes. This trend suggests that when the model's code generation capability is still limited, it learns and leverages self-correct more frequently. As RL training, improves in producing correct code directly, thereby reducing the necessity for correction. This phenomenon reflects an emergent behavior: self-correction is prevalent at the outset but becomes less necessary as the model gains competence.

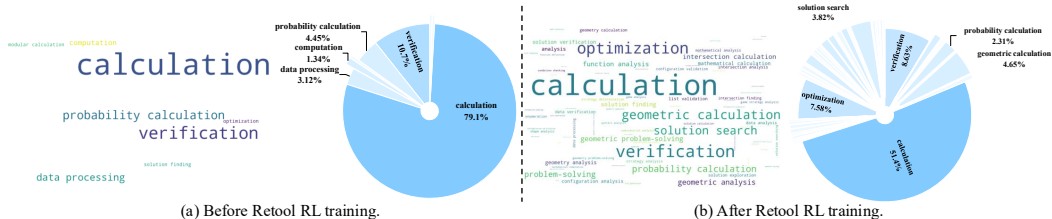

(a) Before Retool RL training.          (b) After Retool RL training.

Figure 5: Code purpose analysis.

- **Code Pass Rate** (Figure 4 (e)): The CI-powered reasoning process involves generating intermediate code that may initially be incorrect, followed by iterative refinement based on interpreter feedback to produce executable code, so we report the average pass rate of last code in incorrect responses. Our analysis reveals that the code pass rate for correct responses remains consistently high, approaching 100%, while the code pass rate for incorrect responses exhibits a declining trend. This pattern suggests that code executability impacts the reasoning process and final result.

- **Code Invocation Timing** (Figure 4 (f)): We also calculate the code invocation timing, which is determined by dividing the start position of code by the total length of the response. This metric reflects the timing of code invocation within the response. The results show that the code invocation timing advances during the RL training process, indicating that the model learns to determine the timing for tool usage.

**Code Purpose Analysis.** We also analysis the differences in code purposes before and after RL training, which reflects the types of code. We employ Claude4-Sonnet (Anthropic, 2025) to classify the primary purpose of code snippets based on their contextual information, then compute the frequency of code purposes that appear more than once, and the results are depicted in Figure 5. The word clouds reveal that calculation and verification are the dominant purposes of code in CI-powered reasoning. After RL training, the code purposes in our model become more diverse, which demonstrates the metacognitive development of adaptive tool selection and enhances the generalizability of ReTool to a broader range of problems.

## 4 BACKGROUND AND RELATED WORK

### 4.1 LLM REASONING

Recent advancements in large language models (LLMs) (Wei et al., 2022; Yao et al., 2023; Luong et al., 2024; OpenAI et al., 2024; Team, 2025a; DeepSeek-AI et al., 2025; xAI, 2023; Claude, 2025; Team et al., 2023; Yang et al., 2024a) indicate significant progress toward cognitive abilities similar to human metacognition through Chain-of-Thought (CoT) prompting, which enhances the reasoning capabilities of LLMs by leveraging step-by-step natural language reasoning Wei et al. (2023); Wang et al. (2023a); Wang & Zhou (2024); Huang et al. (2025); Chen et al. (2025a). Building upon this foundation, recent research has shifted focus from train-time scaling to test-time scaling (Snell et al., 2024), where additional computational resources are allocated during inference to enable the generation of intermediate reasoning steps. Techniques such as stepwise preference optimization (Lai et al., 2024), Monte Carlo Tree Search (MCTS) (Xie et al., 2024), and reinforcement learning (Luong et al., 2024; DeepSeek-AI et al., 2025) have been employed to improve multi-step and long-form mathematical reasoning. Complementing CoT, Program-of-Thought (PoT) reasoning, introduced by Chen et al. (2023) and Gao et al. (2023), integrates external computational tools to simplify and validate complex reasoning steps, resulting in enhanced accuracy.

### 4.2 TOOL INTEGRATED REASONING

Tool-integrated reasoning was first introduced to help LLMs solve computationally intensive mathematical problems with the integration of programming strategies (Chen et al., 2023; Yue et al., 2023; Jin et al., 2025; Song et al., 2025; Wang et al., 2024). Building on this line of work, Wang et al. (2023b) introduced an iterative framework that integrates textual reasoning with code execution, while Chen et al. (2025b) further advanced this direction by applying supervised fine-tuning

on self-curated code-integrated CoT data. However, this approach is inherently limited by its reliance on the specific data distribution, and cannot learn adaptive strategies for tool use—such as determining when and how to invoke tools—through reinforcement learning (RL). A concurrent work (Li et al., 2025b) applied RL to learn tool usage strategies on Qwen2.5-Math models (Yang et al., 2024b) at 1.5B and 7B scales, but the performance remained suboptimal. We further scale up this line of research and propose ReTool, a framework that leverages RL to strategically determine when and how to invoke the code interpreter. Our method outperforms Qwen-Math-72B-TIR (Yang et al., 2024b) and o1-preview (OpenAI, 2024) significantly on AIME2024 and AIME2025. We also present a comprehensive analysis of the learned tool-use behaviors and highlight several key findings regarding the model's cognitive patterns in code invocation after ReTool training.

## 5 CONCLUSION

In this paper, we propose ReTool, a novel reinforcement learning framework that empowers LLMs to self-enhance their mathematical reasoning capabilities through effective Code Interpreter utilization. Our comprehensive experiments on various benchmarks demonstrate that ReTool not only achieves superior accuracy compared to conventional text-based RL approaches, but also converges with significantly fewer training steps. Further analyses also demonstrate generalization to broader tool-use scenarios and emergent behaviors such as code self-correction. Through careful data curation and our specialized tool-using pipeline, ReTool enables models to develop sophisticated computational intervention strategies, paving the way for more efficient and powerful tool-augmented reasoning in LLMs.

### ETHICS STATEMENT

This work does not involve human subjects, personal data, or sensitive information. All experiments were conducted on publicly available datasets in mathematics and STEM domains (e.g., AIME, GSM8K, MATH, GPQA). The datasets were used strictly for research purposes and do not contain identifiable private information. Our methods focus on improving the reasoning capabilities of LLMS by integrating reinforcement learning with external computational tools. We affirm compliance with the ICLR Code of Ethics and confirm that no part of this research raises concerns regarding privacy, discrimination, or conflict of interest.

### REPRODUCIBILITY STATEMENT

We have made every effort to ensure the reproducibility of our work. The full methodological details of ReTool, including cold-start data construction, supervised fine-tuning, reinforcement learning setup, reward design, and evaluation protocols, are described in Sections 2 and 3 of the main text. Comprehensive experimental results across multiple datasets and ablation studies are reported in Tables 1–2 and Figures 3–5. Additional training details such as optimizer, learning rate, batch size, sequence length, masking strategy, and sandbox design are provided in Section 2.3.2. These details should enable independent researchers to replicate our experiments. And the training data and the evaluation scripts are available in the supplementary material.

## ACKNOWLEDGMENTS

We would like to thank Guang Shi, Mingxuan Wang, Renjie Zheng, Chen Dun, and Yun Jiang for their support on this work.

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

## A  APPENDIX

### A.1  THE USE OF LARGE LANGUAGE MODELS (LLMS)

In preparing this paper, we employed large language models (LLMs) solely as writing assistants. Their role was limited to polishing grammar and improving fluency. They were not used for research ideation, methodological development, data analysis, or result interpretation. All scientific contributions, experimental designs, and findings presented in this work are entirely the result of the authors' own efforts.

### A.2  RESULTS ON OTHER BASE MODEL

To further assess the effectiveness of ReTool, we conduct experiments using a code-oriented backbone, Qwen2.5-Coder-7B-Instruct (Hui et al., 2024). Following the same evaluation protocol as in the main results, we report the overall average accuracy on AIME2024, AIME2025, GSM8K, and MATH500. The results are presented in Table 2. We observe that applying our ReTool framework—through both supervised fine-tuning (SFT) and reinforcement learning (RL)—consistently yields substantial improvements across all benchmarks when built on top of Qwen2.5-Coder-7B-Instruct. These findings highlight the generalizability and effectiveness of ReTool across diverse base models.

Table 2: Results on different base model.

| Model | AIME2024 | AIME2025 | GSM8K | MATH500 |
|---|---|---|---|---|
| Qwen2.5-Coder-7B-Instruct | 10.0 | - | 86.7 | 66.8 |
| + ReTool-SFT | 14.69 | 17.29 | 90.67 | 77.2 |
| + ReTool-RL | 46.04 | 32.60 | 92.08 | 84.55 |

### A.3  THE PRACTICAL HELPFULNESS OF THE GENERATED CODE TOWARD THE FINAL SOLUTION.

To more rigorously evaluate the practical usefulness of generated code for solving the target tasks, we conducted a focused analysis of code snippets produced by our models before and after RL training. Specifically, we extracted all code snippets along with their corresponding contexts from the AIME2024 results and asked Claude4-Sonnet (Anthropic, 2025) to assess whether each snippet contributed meaningfully to solving the overall problem. The analysis reveals that, prior to RL training, 406 out of 471 code snippets (86.20%) were judged to be helpful, while 65 (13.80%) were not. After RL fine-tuning, the proportion of helpful code increased: 4,468 out of 5,010 snippets (89.18%) were rated as helpful, compared with 542 (10.82%) deemed unhelpful. These findings indicate not only a consistently high proportion of practically useful code but also a further improvement in code helpfulness following RL training.

### A.4  CI-POWERED REASONING VS. TEXT-BASED REASONING.

We present a case study to illustrate the distinction between CI-powered reasoning after reinforcement learning (RL) training and conventional text-based reasoning prior to RL training, as illustrated in Figure 7 in the Appendix. When faced with the same question, text-based reasoning relies on a "laborious" text-only calculation process, which is prone to numerical errors and often results in incorrect inference outcomes. In contrast, CI-powered reasoning substitutes this complex calculation process with concise code. This approach not only ensures computational accuracy through the assistance of an external code interpreter but also enables the model to focus more effectively on holistic reasoning strategies.

---

**Question**: From an unlimited supply of 1-cent coins, 10-cent coins, and 25-cent coins, … Find the number of values of $N$ between $1$ and $1000$ inclusive for which the greedy algorithm succeeds.

---

**Response**: Alright, let's tackle this problem step by step using code to ensure accuracy. The goal is to determine for each value of $( N )$ from 1 to 1000 whether the greedy algorithm produces the minimum number of coins. If it does, we count that $( N )$. …

```python
def test_specific_N(N_values):
  for N in N_values:
    greedy_result = greedy(N)
    ...
test_results = test_specific_N(N_values_to_test)
print(test_results)
```

<interpreter>Traceback (most recent call last):
File "/tmp/tmpbr3uj7zz/tmpdmj39k6o.py", line 11, in <module> test_results = test_specific_N(N_values_to_test)
File "/tmp/tmpbr3uj7zz/tmpdmj39k6o.py", line 5, in test_specific_N greedy_result = greedy(N)
**NameError: name 'greedy' is not defined** </interpreter>

**Oops, the functions need to be defined in the same scope. Let's correct that:**


```python
def greedy(N):
  denominations = [25, 10, 1]
    ...
def test_specific_N(N_values):
  for N in N_values:
    greedy_result = greedy(N)
    ...
test_results = test_specific_N(N_values_to_test)
print(test_results)
```

<interpreter>{42: (9, 6), 25: (1, 1), 10: (1, 1), 1: (1, 1), 1000: (40, 40)} </interpreter>
…
<answer> \boxed{610} </answer>

---

Figure 6: The case of "aha moment" about code self-correction.

**Question**: Let $N$ be the greatest four-digit positive integer with the property that whenever one of its digits is changed to $1$, the resulting number is divisible by $7$. Let $Q$ and $R$ be the quotient and remainder, respectively, when $N$ is divided by $1000$. Find $Q+R$.

**Text-based Reasoning Before RL**: Okay, so I need to find the greatest four-digit number N such that if any one of its digits is replaced by 1, the resulting number is divisible by 7. Then, I have to find Q and R when N is divided by 1000, and then compute Q + R. Alright, let's break this down step by step…
Therefore, each of these equations can be rearranged to:
1. N ≡ - (1 - A)*1000 mod7
2. N ≡ - (1 - B)*100 mod7
3. N ≡ - (1 - C)*10 mod7
4. N ≡ - (1 - D)*1 mod7
Therefore, all four expressions must be equal to N mod7. Therefore, we can set them equal to each other: -(1 - A)*1000 ≡ - (1 - B)*100 ≡ - (1 - C)*10 ≡ - (1 - D)*1 mod…
Wait, let's check B, C, D for A=9:
B=3A -2 mod7=3*9 -2=27 -2=25 mod7=25 -7*3=25-21=4
C=2A -1 mod7=2*9 -1=18 -1=17 mod7=17 -7*2=17-14=3
D=6A -5 mod7=6*9 -5=54 -5=49 mod7=49 -7*7=49-49=0
So, B=4, C=3, D=0. Therefore, N=9430…
Let me compute each of these modulo7:
1. 1000 ≡6 mod7, so 6 + 100*B + 10*C + D ≡0 mod7
But 100*B ≡2*B mod7,
10*C ≡3*C mod7, so:
6 +2B +3C +D ≡0 mod7
2. 1000*A ≡6A mod7, 100≡2 mod7,
so:
6A +2 +3C +D ≡0 mod7
3. 1000*A +2B +3 +D ≡0 mod7 (since 100*B≡2B, 10*1≡3, 10*C≡3C)
Wait, no. Let me recompute each equation properly.
Equation1 : 1BCD ≡0 mod7
Which is 1000*1 + 100*B + 10*C + D ≡0 mod7 1000≡6, 100≡2, 10≡3, …
Let me tabulate A from1 to9:
A | B=5-4A mod7 | C=6-5A mod7 | D=2-A mod7 | Valid?
1 | 5-4=1 mod7=1 |6-5=1 mod7=1 |2-1=1 |Yes…
**Result**:
- \( N = 5624 \)
- Dividing by 1000 gives \( Q = 5 \) and \( R = 624 \)
- \( Q + R = 5 + 624 = 629 \)
\[ \boxed{629} \]

**CI-powered Reasoning After RL**: Alright, let's tackle this problem step by step. The goal is to find the greatest four-digit positive integer \( N \) such that changing any one of its digits to 1 results in a number divisible by 7. Then, we need to find \( Q + R \) where \( Q \) and \( R \) are the quotient and remainder when \( N \) is divided by 1000…
First, let's compute the values of \( 1000 \mod 7 \), \( 100 \mod 7 \), \( 10 \mod 7 \), and \( 1 \mod 7 \):

```python
# Compute modular values
mod_1000 = 1000 % 7
mod_100 = 100 % 7
mod_10 = 10 % 7
mod_1 = 1 % 7
...
print(f"1 mod 7 = {mod_1}")
```

<interpreter>1000 mod 7 = 6 100 mod 7 = 2 10 mod 7 = 3 1 mod 7 = 1</interpreter>
…
We need to solve these equations for digits \( a, b, c, d \) where \( a \) is from 1 to 9 and \( b, c, d \) are from 0 to 9. Let's use code to iterate through possible values and find the valid \( N \).

```python
def find_valid_n():
    for a in range(9, 0, -1):
        ...
        return N
...
N = find_valid_n()
Q = N // 1000
R = N % 1000
result = Q + R
print(f"N = {N}, Q = {Q}, R = {R}, Q + R = {result}")
```

<interpreter>N = 5694, Q = 5, R = 694, Q + R = 699 </interpreter> …
Thus, the final answer is confirmed to be 699.
<answer> \boxed{699} </answer>

Figure 7: Case of CI-powered reasoning vs. Text-based reasoning.

---

**Template prompt for Reinforcement Learning Rollout**

Solve the following problem step by step. You now have the ability to selectively write executable Python code to enhance your reasoning process. The Python code will be executed by an external sandbox, and the output (wrapped in <interpreter>output</interpreter>) can be returned to aid your reasoning and help you arrive at the final answer. The Python code should be complete scripts, including necessary imports.

**Code Format:**
Each code snippet is wrapped with

```python
 code snippet
```


**Answer Format:**
The last part of your response should be:
<answer>\boxed{'The final answer goes here.'}</answer>

**User Question:**
{question}

**Assistant:**

Figure 8: Template prompt for ReTool rollout.

---

**Template Prompt for Data Curation**

---

You are a helpful AI assistant. Initially, when solving a question, you would need to think step by step, without the ability to use code for calculation. Now, you have the capability to write code to use the code interpreter for calculation. The code will be executed by a sandbox, and the result can be returned to enhance your reasoning process. You can now leverage code to enhance your calculation while still maintaining the reasoning process.

The thinking process can have multiple code snippets. Each code snippet is wrapped with:

```python
code snippet
```
, and should be executable. The returned result is wrapped with <interpreter> execution results \texttt{</interpreter>}.

**Goal:**
Modify the original thinking process to make it more accurate by replacing manual calculation steps that can benefit from code execution with the corresponding code snippets and their interpreter's execution results. The core reasoning logic from the original thinking process, including any unsuccessful attempts, should remain unchanged. You should only replace the necessary manual calculation steps with code and interpreter's execution results, without altering the rest tokens of the thinking process. Wrap the revised thinking process within <revised_thinking_process> and </revised_thinking_process>}.

**User Question:**
{question}

**Original Thinking Process (without code interpreter's support):**
<original_thinking_process> {original_response} </original_thinking_process>

**Details:**
1. Identify sections where code execution could speed up the reasoning process or make the calculation more accurate.
2. Replace the manual calculation steps with code snippets and the corresponding interpreter's execution results.
3. Keep the logical flow of the reasoning process intact, including any failed exploration attempts that were part of the initial process.
4. The code snippets should be complete scripts, including necessary imports, and should not contain markdown symbols like

```python
code snippet
```
.
5. Outputs in the code snippets must explicitly call the **print** function.
6. Execution results should match the model's output exactly, with no extra or missing tokens.
7. If the Original Thinking Process does not include an <answer> section at the end, please add it in the Revised Thinking Process:
<answer> \boxed{'The final answer goes here.'} </answer>

**Revised Thinking Process (With code interpreter's support):**

---

Figure 9: Template prompt for data curation.

