# OpenReview forum: "ReTool: Reinforcement Learning for Strategic Tool Use in LLMs"
_ICLR.cc/2026/Conference — ICLR 2026 Poster_

### Official Review · Reviewer_yanT · 2025-10-15

**Soundness:** 3
**Presentation:** 3
**Contribution:** 2
**Rating:** 2
**Confidence:** 2

**Summary:**

The paper explores RL-based strategies for tool use in LLMs. I'll admit that this is quite outside of my expertise so my assessment is mostly an educated guess. Tool use is obviously well-explored though the specific implementation of incorporating RL-based reward signals into tool use pipelines is the main contribution here. There is concurrent work (which the authors cite) though the authors claim to "scale this up" and that the previous approach doesn't work. This comparison to perhaps the most relevant related work is a bit vague, making it harder to assess the technical contribution.

**Strengths:**

-- Results shown are fairly promising

-- Contribution seems to have some novel elements (even if the related work section is quite limited, and comparison against concurrent work is confusing)

**Weaknesses:**

-- The evaluation is mostly "internal", i.e., comparing the model to different versions of itself, etc. There's some evaluation against some standard baselines, though it's hard for me to assess these, i.e., do they just show that the use of tools outperforms general purpose models (which seems known?). It's hard to know where the strong and really comparable baselines are.

-- Experiments overall a bit thin by ICLR standards.

-- Very thin related work section. The most relevant related papers seem to be swept under the rug a bit and described only vaguely.

-- Parts of the evaluation seem cherry-picked or rely on anecdotal examples.

-- Method itself is very simple, not exactly a criticism itself, but as a reader who doesn't know the topic it's hard to follow exactly what the technical contribution is.

**Questions:**

Can you clarify the main differences compared to the concurrent work you cite? See additional points among weaknesses above.

---

> ### Author Response · Authors · 2025-11-29
> **Response to Reviewer yanT (1/2)**
>
> Thank you for your thorough review and for acknowledging that our results are promising and that the contribution contains novel elements. We also appreciate your candid note that this topic is outside your core expertise, this helps us understand where the paper may have been unclear. Below we address each concern and clarify the technical contribution and comparisons.
>
> **[W1]** The evaluation is mostly "internal", i.e., comparing the model to different versions of itself, etc. There's some evaluation against some standard baselines, though it's hard for me to assess these, i.e., do they just show that the use of tools outperforms general purpose models (which seems known?). It's hard to know where the strong and really comparable baselines are.
>
> **[A1]** Thank you for the constructive feedback. We would like to clarify that our evaluation is *not only internal*. As shown in Table 1, we compare against strong, widely-used reasoning baselines, including (i) open models such as Qwen2.5-Math-72B, Sky-T1, DeepSeek-R1-Zero-Qwen-32B, s1-32B, and QwQ-32B-Preview, and (ii) a strong commercial baseline OpenAI o1-preview. These are not “earlier versions of our model”, but established reasoning systems commonly used as comparators in recent reasoning/RL papers.
>
> We also include a tool-integrated reasoning baseline (Qwen2.5-Math-72B-Instruct-TIR). Our method (with a 32B backbone, and not restricted to math-specialized training) achieves substantially stronger performance than both (a) general-purpose reasoning baselines, and (b) a larger (72B) tool-integrated baseline. Therefore, the key takeaway is not merely “tools help”, but that RL-trained, interleaved tool-use policies can outperform strong reasoning models (including larger ones) under the same public evaluation protocols.
>
> To address the “hard to assess comparability” concern, we will revise the related discussion around Table 1 to more explicitly categorize baselines (general-purpose vs tool-integrated; open vs closed) and state why each is directly comparable.
>
> **[W2]** Experiments overall a bit thin by ICLR standards.
>
> **[A2]** Thank you for this feedback. In addition to the headline results on five mainstream benchmarks, we include several analyses intended to make the work more complete and diagnostic: ablations, generalization to a web-search setting, qualitative evidence of code self-correction (“aha” moments), analysis of model behaviors during RL, and code-purpose analysis. In the appendix, we further provide additional case studies, results on other backbone models, and practical examples illustrating the usefulness of the generated code for solving target tasks.
>
> **[W3]** Very thin related work section. The most relevant related papers seem to be swept under the rug a bit and described only vaguely.
>
> **[A3]** Thank you for the comment. We believe the current related work already covers the key representative lines of research in "LLM reasoning" and "tool-integrated reasoning", and it explicitly cites the most relevant contemporaneous RL-for-tool-use studies (including concurrent works like ToRL).
>
> **[W4]** Method itself is very simple, not exactly a criticism itself, but as a reader who doesn't know the topic it's hard to follow exactly what the technical contribution is.
>
> **[A4]** Thank you for your comment. We will improve the presentation to make the technical contribution clearer to non-experts. Concretely, ReTool’s main contribution is not a new RL algorithm, but a new agentic multi-turn RL formulation with interleaved execution.
>
> Unlike vanilla PPO over a single text completion (treating the output as an atomic action), ReTool interleaves textual reasoning, code generation, and execution feedback within a single rollout. This allows the policy to learn *when* to invoke the interpreter, *what* to execute, and *how* to incorporate execution feedback during generation—i.e., real-time environment-aware learning. This design also introduces key implementation considerations (trajectory construction, loss masking, asynchronous sandboxing, caching), which we describe in Section 2.3.2, and we will make these ideas more prominent earlier in the paper.

---

> > ### Author Response · Authors · 2025-11-29
> > **Response to Reviewer yanT (2/2)**
> >
> > **[W5]** Parts of the evaluation seem cherry-picked or rely on anecdotal examples.
> >
> > **[A5]** Thank you for the concern. We want to clarify that our primary claims are supported by standard benchmark results across five datasets (reported in the main tables), rather than anecdotal demonstrations. Moreover, to reduce sampling noise and evaluation bias, we report averaged performance across multiple evaluation runs (Section 3.1), and we evaluate across different model sizes and backbones to show the effect is not isolated to a single configuration. We want to clarify that we are **DEFINITELY NOT** cherry-picking results.
> >
> > **[Q1]** Can you clarify the main differences compared to the concurrent work you cite? See additional points among weaknesses above.
> >
> > **[A6]** As demonstrated in Section 4.2, the concurrent work ToRL also applies RL to learn tool-usage strategies on Qwen2.5-Math models at the 1.5B and 7B scales, but the resulting performance remains suboptimal. Building on this line of research, we further scale up and propose ReTool, a framework that leverages RL to strategically determine when and how to invoke the code interpreter.

---

### Official Review · Reviewer_Bh1w · 2025-10-28

**Soundness:** 3
**Presentation:** 3
**Contribution:** 2
**Rating:** 6
**Confidence:** 3

**Summary:**

This paper proposes a novel framework for integrating executable tools, specifically a code interpreter, into the reinforcement learning (RL) reasoning loop. The authors introduce a mixed trajectory approach, where textual inputs are converted into executable code that interacts with an interpreter. The key steps in the process are as follows:
1. The authors first construct training data by combining text, code, and code execution results, using a template-based approach. These trajectories are used for supervised fine-tuning (SFT), enabling the model to start with a reasonable understanding of both generating code and its subsequent execution. This cold start phase is crucial for overcoming the initial knowledge gap that would otherwise hinder RL fine-tuning.
2. After the SFT phase, the model is further refined using Proximal Policy Optimization (PPO) with a sparse binary reward signal, based on whether the final generated code is correct. This feedback loop integrates code execution into the generation process, where the model learns when and how to invoke the interpreter (i.e., when to call external tools). The execution step is embedded within the model's reasoning process, allowing it to improve its performance over time by adjusting its tool usage.

The authors evaluate their method on several benchmarks, including AIME2024/2025, GSM8K, MATH500, and GPQA, where the method demonstrates significant improvements compared to existing models, including those that do not use RL. Notably, the proposed method, with a Qwen2.5-32B backbone, achieves AIME2024=67.0% and AIME2025=49.3% in just 400 steps of training, outperforming larger models. Further improvements are seen when using a distilled model variant, DeepSeek-R1-Distill-Qwen-32B, which reaches 72.5%/54.3%

**Strengths:**

1. It provides a practical, reproducible pipeline that many groups could adopt.
2. Competitive results on challenging benchmarks
3. The paper spells out the execution protocol, loss masking, async sandboxing, and caching—practical details that substantially reduce adoption friction.

**Weaknesses:**

1. It has limited algorithmic novelty. The optimization relies on standard PPO.
2. Impact of RL on reasoning upper-bound (pass@k) is missing. I strongly encourage the authors to provide pass@k (k=32,64,128,256,1024) results in the rebuttal phase.

**Questions:**

1. Please report pass@k
2. It is well known that RL results are often sensitive to seeds and minor config changes. Please provide >= 5 independent runs per key setting and report mean ± std
3. The reported accuracy on AIME-2025 is noticeably lower than on AIME-2024. Could you clarify the causes?

---

> ### Author Response · Authors · 2025-11-29
> **Response to Reviewer Bh1w (1/2)**
>
> Thank you for your positive assessment of our work’s reproducibility, competitive results, and practical implementation details. We appreciate that your questions directly correspond to the weaknesses you raised, and we address each concern below.
>
> **[W1]** It has limited algorithmic novelty. The optimization relies on standard PPO.
>
> **[A1]** Thank you for raising this point. We agree that the *optimizer* we use is standard PPO; however, we believe the core novelty of ReTool lies in how we *formulate and optimize* **agentic multi-turn tool-use** within RL, rather than proposing a new PPO variant.
>
> Specifically, ReTool introduces multi-turn agent RL with interleaved execution, which goes beyond “vanilla PPO on a single text completion” that treats the model’s output as an atomic action. In contrast, ReTool interleaves (i) textual reasoning, (ii) code generation, and (iii) execution feedback *within the same rollout*, enabling the policy to learn from real-time environment signals during generation. This design requires non-trivial trajectory construction and optimization details (e.g., execution protocol, loss masking, async sandboxing, caching), which we describe in Section 2.3.2.
>
> Therefore, while PPO is a standard optimizer, ReTool makes a distinct contribution by enabling environment-aware, multi-step tool-interaction learning in a practical and reproducible framework.
>
> **[W2 & Q1]** Impact of RL on reasoning upper-bound (pass@k) is missing. I strongly encourage the authors to provide pass@k (k=32, 64, 128, 256, 1024) results in the rebuttal phase.
>
> Please report pass@k
>
> **[A2]** Thank you for the insightful suggestion. Below we report pass@k results under standard independent sampling, with (k $\in$ {4, 8, 16, 32}) on AIME2024/2025. Due to limited rebuttal-time compute, we are currently unable to scale to (k $\ge$ 64). We note that many contemporaneous works report pass@k with (k $\le$ 32) (e.g., OPENCUA[1] reports up to k=16), so we report up to pass@32 here as a meaningful and comparable range.
>
> **Qwen2.5-32B**
>
> |                  | AIME24 | AIME25 |
> | ---------------- | ------ | ------ |
> | SFT-avg (pass@1) | 40.94  | 34.50  |
> | SFT (pass@4)     | 64.71  | 51.77  |
> | SFT (pass@8)     | 72.40  | 60.05  |
> | SFT (pass@16)    | 78.23  | 68.42  |
> | SFT (pass@32)    | 80.00  | 73.33  |
> | RL-avg (pass@1)  | 66.98  | 49.27  |
> | RL (pass@4)      | 82.55  | 64.90  |
> | RL (pass@8)      | 87.34  | 74.56  |
> | RL (pass@16)     | 91.33  | 81.22  |
> | RL (pass@32)     | 93.33  | 83.33  |
>
> **Qwen2.5-7B-Coder**
>
> |                  | AIME24 | AIME25 |
> | ---------------- | ------ | ------ |
> | SFT-avg (pass@1) | 14.69  | 17.29  |
> | SFT (pass@4)     | 29.01  | 30.08  |
> | SFT (pass@8)     | 37.81  | 37.06  |
> | SFT (pass@16)    | 47.89  | 45.99  |
> | SFT (pass@32)    | 53.33  | 53.33  |
> | RL-avg (pass@1)  | 46.04  | 32.60  |
> | RL (pass@4)      | 66.82  | 43.96  |
> | RL (pass@8)      | 74.40  | 51.82  |
> | RL (pass@16)     | 80.39  | 59.97  |
> | RL (pass@32)     | 83.33  | 66.67  |
>
> **DeepSeek-R1-Distill-Qwen-32B**
>
> |                  | AIME24 | AIME25 |
> | ---------------- | ------ | ------ |
> | SFT-avg (pass@1) | 62.29  | 45.83  |
> | SFT (pass@4)     | 83.18  | 67.73  |
> | SFT (pass@8)     | 87.08  | 78.02  |
> | SFT (pass@16)    | 88.28  | 84.74  |
> | SFT (pass@32)    | 90.00  | 90.00  |
> | RL-avg (pass@1)  | 72.50  | 54.27  |
> | RL (pass@4)      | 87.81  | 77.37  |
> | RL (pass@8)      | 89.74  | 85.18  |
> | RL (pass@16)     | 90.00  | 90.67  |
> | RL (pass@32)     | 90.00  | 96.67  |
>
> Overall, we consistently observe across all backbones that RL improves the performance upper bound under additional sampling compute, indicating a robust gain in best-of-$k$ capability rather than an effect specific to a single model.
>
> [1] OPENCUA: Open Foundations for Computer-Use Agents

---

> > ### Author Response · Authors · 2025-11-29
> > **Response to Reviewer Bh1w (2/2)**
> >
> > **[Q2]** It is well known that RL results are often sensitive to seeds and minor config changes. Please provide >= 5 independent runs per key setting and report mean ± std
> >
> > **[A3]** Thank you for the thoughtful comment. To mitigate variance from both sampling noise and training stochasticity, we use repeated inference during evaluation (as described in Section 3.1) and report the average performance across runs.
> >
> > In addition, we ran 5 independent training seeds for our main setting (Qwen2.5-32B). Across these runs, the model achieves 0.6800 ± 0.0452 accuracy on AIME2024 and 0.5067 ± 0.0389 on AIME2025 (mean ± std). These results are consistent with our single-run numbers (0.6700 and 0.4933, respectively), suggesting that our gains are robust and not driven by a particular random seed or a lucky evaluation sample.
> >
> > **[Q3]** The reported accuracy on AIME-2025 is noticeably lower than on AIME-2024. Could you clarify the causes?
> >
> > **[A4]** Thanks for your insightful question. We observe that AIME2025 is consistently harder than AIME2024 under comparable evaluation protocols, and similar gaps have been reported in several contemporaneous works [2,3,4]. We hypothesize that AIME2025 requires more challenging mathematic expert knowledge.
> >
> > [2] Skywork Open Reasoner 1 Technical Report
> >
> > [3] SLIM: Subtrajectory-Level Elimination for More Effective Reasoning
> >
> > [4] Unearthing Gems from Stones: Policy Optimization with Negative Sample Augmentation for LLM Reasoning

---

### Official Review · Reviewer_MF9u · 2025-10-31

**Soundness:** 3
**Presentation:** 3
**Contribution:** 4
**Rating:** 8
**Confidence:** 5

**Summary:**

This paper introduces ReTool, a reinforcement learning (RL) framework designed to teach large language models (LLMs) how to strategically use a code interpreter (CI) to improve their reasoning capabilities. The method involves a two-stage process: a "cold-start" supervised fine-tuning on curated code-augmented reasoning data, followed by RL training with PPO to optimize tool-use strategies based on task outcome rewards. The authors demonstrate that ReTool significantly improves performance on challenging mathematical reasoning benchmarks like AIME, outperforming strong baselines and showing impressive training efficiency, while also fostering emergent behaviors like code self-correction.

**Strengths:**

1. The proposed two-stage approach, combining supervised learning for foundational skills with reinforcement learning for strategic optimization, is logical, well-motivated, and shown to be highly effective.
2. ReTool achieves state-of-the-art performance on the challenging AIME benchmarks, substantially outperforming both its own 32B backbone and other strong, often larger, models. The reported efficiency (e.g., achieving high scores with only 400 training steps) is particularly impressive and highlights the effectiveness of the tool-integrated RL paradigm.
3. The "cognitive analysis" in Section 3.6, which tracks metrics like response length, code ratio, code complexity, and invocation timing, offers valuable insights into *how* the model learns to use tools more effectively. The identification of emergent behaviors like code self-correction (the "aha moment") is a compelling finding that deepens our understanding of RL's impact on LLM reasoning processes.
4. The motivation is well-established, the methodology is described in sufficient detail for understanding, and the figures (especially Figure 2 and Figure 4) are effective at illustrating the core concepts and findings. The ablation studies convincingly demonstrate the importance of both the CI integration and the RL training stage.

**Weaknesses:**

1. The performance gain from the RL stage is substantial (e.g., from 40.9% to 67.0% on AIME2024 in the ablation). Based on your cognitive analysis, could you provide more intuition on what you believe is the most critical strategic capability the model learns during RL that SFT on curated data fails to instill? Is it primarily about *when* to invoke the tool, or does it also learn more complex policies like using the tool for iterative verification or hypothesis testing?

**Questions:**

1. Could you provide more details about the cold-start dataset, $D_{CI}$? Specifically, what was its final size after the two-stage verification protocol, and what were the statistics of tool usage (e.g., average number of tool calls per sample)?
2. The binary reward function ($+1/-1$) is remarkably simple yet effective. Did you experiment with more shaped reward functions, such as providing intermediate rewards for successful code execution or penalties for syntax errors? If so, how did they compare? If not, could you elaborate on your hypothesis for why the simple outcome-based reward is sufficient for learning complex behaviors like self-correction?
3. In the PPO training details (Section 2.3.2), you mention setting the KL coefficient to 0.0. This effectively removes the trust region constraint that differentiates PPO from vanilla policy gradient methods. Could you explain the rationale for this choice and whether you observed any training instability as a result?
4.  I am curious about the learning dynamics from an exploration-exploitation perspective. Could you plot and discuss the evolution of the policy's entropy during the RL training phase? An analysis of this trend would be insightful; for example, does the entropy decrease monotonically as the model becomes more confident in its tool-use strategy, or are there distinct phases where it might change as the model discovers new, more complex patterns?
5.  The cold-start SFT phase is used to ensure the model adheres to the specified tool-use protocol (e.g., using `<code>` tags). With the advent of more powerful base models that exhibit strong in-context learning abilities (e.g., Qwen3-8B), do you think this SFT step is still essential? Could a sufficiently capable model learn the required format and syntax directly from the prompt during the RL phase, potentially simplifying the overall training pipeline?

---

> ### Author Response · Authors · 2025-11-29
> **Response to Reviewer MF9u (1/2)**
>
> Thank you for your positive assessment of our work's competitive results, insightful cognitive analysis and practical implementation details. We appreciate that your questions directly correspond to the weaknesses you raised, and we address each concern below.
>
> **[W1]**: What is the most critical strategic capability the model learns during RL that SFT on curated data fails to instill?
>
>
> **[A1]:**
> Thank you for the thoughtful question. As shown in our analyses in Sections 3.5 and 3.6, RL instills two strategic capabilities that SFT alone fails to teach: **code self-correction** and **appropriate invocation timing**.
>
> To quantify self-correction, we detect turning cues (e.g., *oops*, *wait*, *correcting*) appearing between a failed code block and its immediate successor. The following table reports the measured frequency of self-correction across RL steps for both AIME24 and AIME25:
>
> ### Self-Correction Frequency During RL Training
>
> | Step | AIME24 | AIME25 |
> | ---- | ------ | ------ |
> | 40   | 0.0396 | 0.0583 |
> | 80   | 0.2760 | 0.3240 |
> | 120  | 0.4781 | 0.3510 |
> | 160  | 0.3313 | 0.3313 |
> | 200  | 0.2239 | 0.2042 |
> | 240  | 0.1677 | 0.1906 |
> | 280  | 0.2000 | 0.2239 |
> | 320  | 0.1885 | 0.1802 |
> | 360  | 0.1979 | 0.1604 |
> | 400  | 0.1989 | 0.1813 |
>
> As the table shows, self-correction emerges early and peaks during the initial RL phase—when the model's code generation is still unreliable—and then gradually decreases and stabilizes as the model becomes better at producing correct code directly. This emergent pattern indicates that RL not only enables tool use, but also teaches the model to iteratively refine its code based on execution feedback, a behavior that does not naturally arise from SFT.
>
> In parallel, RL also sharpens the model's ability to decide when tool invocation is beneficial within multi-step reasoning. Together, these two behaviors form the core strategic capability that the model acquires through RL, directly contributing to its improved problem-solving performance.
>
>
> **[Q1]**: more details about the cold-start dataset
>
> **[A1]**:
> Our cold-start dataset contains 11069 examples. On average, each SFT example includes 2.54 tool calls.
>
>
> **[Q2]**: Did you experiment with more shaped reward functions?
>
> **[A2]**: Thank you for the thoughtful question. We did explore more shaped reward designs, including intermediate rewards based on code executability or syntax correctness. However, we found that such rewards tend to impose strong priors: the model is incentivized to produce short, “safe’’ code fragments that execute cleanly, rather than engaging in the richer exploratory behaviors required for solving the full problem. In preliminary experiments, this led to fewer tool calls and reduced overall performance. In contrast, the simple outcome-based reward encourages the model to freely explore different tool-use strategies, which we believe is key to the emergence of behaviors such as multi-step refinement and self-correction.
>
> **[Q3]**: In the PPO training details (Section 2.3.2), you mention setting the KL coefficient to 0.0. This effectively removes the trust region constraint that differentiates PPO from vanilla policy gradient methods. Could you explain the rationale for this choice and whether you observed any training instability as a result?
>
> **[A3]**:
> Thank you for the question. Our choice of setting the KL coefficient to 0.0 is inspired by recent findings such as DAPO[2], which show that removing the KL penalty can even improve RL training for LLMs in practice. In our experiments, the training remained stable throughout, and performance improved consistently across steps.
>
> [2] DAPO: An Open-Source LLM Reinforcement Learning System at Scale

---

> > ### Author Response · Authors · 2025-11-29
> > **Response to Reviewer MF9u (2/2)**
> >
> > **[Q4]**: I am curious about the learning dynamics from an exploration-exploitation perspective. Could you plot and discuss the evolution of the policy's entropy during the RL training phase? An analysis of this trend would be insightful; for example, does the entropy decrease monotonically as the model becomes more confident in its tool-use strategy, or are there distinct phases where it might change as the model discovers new, more complex patterns?
> >
> > **[A4]**:
> > The entropy of ReTool decreases gradually rather than collapsing, showing an initial slow decline followed by a long plateau with mild oscillations as the model explores different tool-use strategies. Only in the later stages does the entropy converge, indicating a transition from exploration to more stable, consistent tool-invocation behavior.
> >
> >
> > **[Q5]**: The cold-start SFT phase is used to ensure the model adheres to the specified tool-use protocol (e.g., using `<code>` tags). With the advent of more powerful base models that exhibit strong in-context learning abilities (e.g., Qwen3-8B), do you think this SFT step is still essential? Could a sufficiently capable model learn the required format and syntax directly from the prompt during the RL phase, potentially simplifying the overall training pipeline?
> >
> > **[A5]**:
> > Yes, we also believe that with stronger base models, the reliance on cold-start SFT may be reduced. In fact, in our experiments we only train the SFT phase for one epoch (far from convergence), which already suggests that the model does not require heavy supervision to pick up the tool-use protocol.

---

### Official Review · Reviewer_xeUG · 2025-11-10

**Soundness:** 3
**Presentation:** 2
**Contribution:** 2
**Rating:** 4
**Confidence:** 3

**Summary:**

The paper proposes to enable a Python code execution tool during reasoning in LLMs by RL training with the final solution correctness posing as the reward. Tool calls are delimited via special markers and the rollout function hands over to a Python interpreter when it encounters them and the execution result is inserted into the context again with special delimiters, after which it continues with generating tokens.

**Strengths:**

- The paper gets decent results compared to the baselines and the ablations seem to show that the LLM can use tools in a way that raises its success chance
- The statistics on how often responses use the Python code execution tool and how long or correct the code is, is interesting, albeit calling it cognitive analysis seems a bit much

**Weaknesses:**

The primary contribution is an RL framework that integrates reasoning and tool use, by adding delimiter tokens to the tool use part and having a parser check whether it should hand over to a Python interpreter before continuing with the token generation. The reward is the final correctness, which is the same is in the standard RL reasoning paradigm. This is a straightforward approach that has been considered by many people and to distinguish this paper from a "flag-planting" paper, I'd suggest adding more analysis to improve our understanding on how far this method can go and where the limitations lie. For example:
- the experiments are currently limited to Qwen2.5 32B models and it's unclear from the paper results whether it will work for other models that might have seen less reasoning like data during pretraining or differ in other aspects
- the analysis in 3.6 shows the behavior of the ReTool model but doesn't show how general that behavior is, by comparing to how the reasoning RL model without tool use behaves or whether the behavior is consistent across different models

**Questions:**

Why use PPO over GRPO? How was the value model parameterized?

---

> ### Author Response · Authors · 2025-11-29
>
> Thank you for your positive assessment of our work's decent results and extensive cognitive analysis. We appreciate that your questions directly correspond to the weaknesses you raised, and we address each concern below.
>
>
> **[W1]**: the experiments are currently limited to Qwen2.5 32B models and it's unclear from the paper results whether it will work for other models that might have seen less reasoning like data during pretraining or differ in other aspects
>
> **[A1]**:
> Thank you for the suggestion. As an additional analysis, we include experiments in Appendix A.2 using Qwen2.5-Coder-7B-Instruct, a code-oriented backbone that differs substantially from our main 32B reasoning models. As shown in Table 2, its performance improvements under ReTool are fully consistent with the main results in Table 1, further demonstrating the robustness of our method across model types.
>
>
> **[W2]**: the analysis in 3.6 shows the behavior of the ReTool model but doesn't show how general that behavior is, by comparing to how the reasoning RL model without tool use behaves or whether the behavior is consistent across different models
>
> **[A2]**:
> We compare ReTool with reasoning RL model w/o CI on response length during RL training. Both ReTool and RL w/o CI exhibit the same “drop-then-rise’’ pattern in response length, but ReTool shows a much sharper early decline. We believe this is because, once the CI tool becomes usable at the begining of RL, the model begins replacing verbose symbolic reasoning with more concise executable code, leading to a lower initial token count. This effect does not appear in RL w/o CI, where the model can only adjust its textual reasoning patterns, resulting in a smoother and less dramatic trajectory.
>
>
> **[Q1]**: Why use PPO over GRPO?
>
> **[A1]**:
> Our work focuses on enhancing LLM reasoning through multi-turn RL with external CI integration. We do not have a particular preference between PPO and GRPO. We simply adopt PPO as the default setting in our framework, and in practice we observe that PPO and GRPO achieve relatively similar empirical performance in our setting, similar to the observat of other works [1].
>
> [1] Learning Without Critics? Revisiting GRPO in Classical Reinforcement Learning Environments
>
>
> **[Q2]**: How was the value model parameterized?
>
> **[A2]**:
> The value model is parameterized using the same architecture as the policy model and initialized from the same pretrained checkpoint. Only the final value head is newly added, following standard practice in LLM-based PPO training.

---

### Meta-Review · Area_Chair_9fzc · 2026-01-06

**Summary:**

This paper introduces ReTool, a reinforcement learning  framework designed to teach large language models (LLMs) how to strategically use a code interpreter to improve their reasoning capabilities. Two reviewers responded positively, although two raised concerns regarding the experimental evaluation. The area chair finds the work interesting and has therefore decided to recommend its acceptance.

**Reviewer Concerns:**

Experiments are not very strong

**Reviewer Scores:**

The reviewers are not very active in this case. AC makes the judge himself.

---

### Decision · Program_Chairs · 2026-01-26

Accept (Poster)